

# Effect of mulching and organic manure on maize yield, water, and nitrogen use efficiency in the Loess Plateau of China

Yingying Xing, Jintao Fu and Xiukang Wang

Key Laboratory of Applied Ecology of Loess Plateau, College of Life Science, Yan'an University, Yan'an, Shaanxi, China

## ABSTRACT

Current agricultural practices prioritize intensive food production, often at the expense of environmental sustainability. This approach results in greenhouse gas emissions and groundwater pollution due to over-fertilization. In contrast, organic agriculture promotes a more efficient use of non-renewable energy, improves soil quality, and reduces ecological damage. However, the effects of mulching and organic manure on maize yield, water use efficiency (WUE), and nitrogen use efficiency (NUE) in China's Loess Plateau have not been sufficiently researched. In 2017 and 2018, an experiment utilizing a randomized complete block design with two factors (two mulching levels × three organic nitrogen application rates) was conducted. The water content of the upper soil layer was found to be 12.6% to 19.4% higher than that of the subsoil layer. Across all soil depths and years, the soil nitrate-N content in mulched treatments was 10% to 31.8% greater than in non-mulched treatments with varying organic nitrogen rates. Additionally, mulching resulted in an increase in grain yield of 9.4% in 2017 and 8.9% in 2018 compared to non-mulched treatments. A significant interaction was observed between mulching and organic nitrogen application rate concerning WUE, alongside a negative correlation between WUE and NUE. These findings suggest that the application of 270 kg N ha$^{-1}$ of sheep manure in conjunction with mulching is a highly recommended practice for the Loess Plateau, thereby supporting sustainable agricultural strategies.

## INTRODUCTION

By 2050, global food demand is projected to double compared to the early 21st century (*Wang, 2022*). However, the expansion of farmland and irrigation resources is expected to be minimal. This situation is particularly critical in China, where each of the 1.3 billion people has access to only 0.1 hectares of farmland (*Chen et al., 2019*). To meet this growing demand, it is essential to enhance agricultural productivity on existing land. Mineral fertilizers play a crucial role in China's agriculture (*Li et al., 2024a*), contributing significantly to recent increases in grain yields due to their ease of application and substantial impact (*Wossen et al., 2023*). However, improper and excessive use of these fertilizers has resulted in soil environmental issues, posing a threat to sustainable

Corresponding author
Xiukang Wang,
wangxiukang@yau.edu.cn

agriculture (*Wendimu, Yoseph & Ayalew, 2023*; *Xing & Wang, 2024*). Long-term application of animal manure has been shown to increase nitrogen levels in the topsoil compared to unfertilized controls (*Govednik et al., 2023*). Additionally, prolonged use of mineral fertilizers can lead to increased soil bulk density, reduced field water capacity, soil compaction, and impaired surface soil permeability, all of which can harm farmland ecosystems (*Khan et al., 2022*). In contrast, the application of organic manure can enhance soil nutrients, enzyme activity, microbial activities, and the overall soil ecological environment (*Iqbal et al., 2022*). Despite these insights, there is a lack of comprehensive studies comparing the effects of mineral fertilizers and organic manure on soil properties in the Loess Plateau (*Wang et al., 2019*).

The appropriate use of fertilizers can enhance soil quality, whereas improper application rates can degrade the physical, chemical, and biological properties of the soil (*Jin et al., 2022*). The primary objective of nitrogen (N) fertilizer in agriculture is to supply sufficient nitrogen for optimal crop growth and yield while minimizing environmental impacts (*Espie & Ridgway, 2020*). In China, traditional practices characterized by high mineral fertilizer usage have contributed to a decline in soil quality (*Sofo, Zanella & Ponge, 2022*). A growing emphasis on energy conservation and environmental protection is shifting the focus toward the utilization of locally available organic materials instead of mineral fertilizers (*Manna et al., 2021*). Organic materials such as farmyard manure, compost, poultry manure, crop residues, and green manure are increasingly recognized as valuable nutrients sources (*Raza et al., 2022*). Unlike mineral fertilizers, organic fertilizers release nutrients gradually, thereby ensuring long-term residual effects (*Dutta et al., 2022*). This transition aims to achieve comparable grain yields while reducing reliance on fossil fuels for mineral fertilizer production (*Wei et al., 2020*). The increased recycling of organic materials is anticipated to release substantial nutrients, particularly nitrogen, which will help to lower fertilization costs and sustain crop yields.

The application of organic fertilizers significantly enhances soil organic matter and improves soil aggregate formation (*Tian et al., 2022*). Over the long term, these fertilizers increase the availability of phosphorus and stabilize the carbon-to-nitrogen ratio in the soil (*Xie et al., 2024*). Furthermore, organic fertilization ensures a continuous supply of nutrients throughout the various stages of crop growth, facilitated by increased activity of rhizosphere microorganisms, which enhance nutrient availability and boost yields (*Ayamba et al., 2023*). Additionally, organic fertilizers improve both water and nutrient use efficiency (*Poddar et al., 2022*; *Zhang et al., 2020*).

In dryland agriculture, *in situ* soil and water management is crucial for boosting productivity. Plastic film mulching is one of the most effective methods, significantly reducing soil water evaporation, enhancing soil water storage, optimizing water supply-and-demand between soil and crops, and increasing nutrient availability (*Zhang et al., 2024*). This technique improves water and fertilizer use efficiency by modifying the soil environment, increasing soil temperature, and decreasing evaporation (*Ramos, Darouich & Pereira, 2024*). Mulching significantly enhances soil water content in the 0–30 cm layer, aiding water conservation and higher yields (*Farooq et al., 2019*). It also improves the

lateral and vertical availability of soil water, supporting crop growth and yield (*Gao et al., 2024*). Changes in soil moisture impact soil nitrate-N distribution in the root layer, a key indicator of soil fertility and productivity (*Shen et al., 2023*; *Meng et al., 2024*). While appropriate fertilizer application boosts crop yields, excessive fertilization under water-scarce conditions can reduce yields (*Wen et al., 2024*). Careful nitrogen management is essential to prevent environmental pollution and financial losses. Thus, understanding the relationship between nitrogen fertilizer and the vertical distribution of soil nitrate-N is crucial.

Previous research has quantified nitrogen distribution under plastic film mulching and fertilizer application (*Huang et al., 2024*). However, the effects of combining mulching with varying rates of organic fertilizer on maize yield, as well as water and fertilizer use efficiency in the Loess Plateau, remain underexplored. We hypothesize that the application of mulching and organic manure significantly influences maize grain yield and nitrogen use efficiency. This study aims to investigate the effects of mulching and different rates of organic manure on soil moisture, soil nitrate distribution, maize grain yield, and both water and nitrogen use efficiency in the Loess Plateau of China.

## MATERIALS AND METHODS

### Study site description

The field experiment was carried out in 2017 and 2018 at the Ansai Field Experiment Station, Shaanxi Province, China (36°38′N, 109°10′E; 1,109 m a.s.l.). The climate is mild and semiarid, with an annual average temperature of 9.2 ± 2.1 °C, a monthly average maximum temperature of 21.5 °C (July), and a monthly average minimum temperature of −6.8 °C (January). The average annual sunshine duration of 2,230 h exceeds 171 days without frost. From 1990 to 2018, the annual average precipitation in this region was 582 ± 114 mm. The local soil type is entisols, as defined by the USDA textural classification system. Before the 2017 experiment, the physical and chemical properties of the 0–40 cm soil layer were as follows: pH 8.4; soil bulk density 1.32 g cm$^{-3}$; soil total carbon content 12 g kg$^{-1}$; soil total nitrogen 0.74 g kg$^{-1}$; soil available phosphorus 35.3 mg kg$^{-1}$; soil available potassium 125.3 mg kg$^{-1}$; mineral nitrogen 12.1 mg kg$^{-1}$.

The total rainfall in the maize growing season was 287 and 351 mm in 2017 and 2018, respectively (Fig. 1). In June, more rainfall was observed in 2018 (60.4 mm) than in 2017 (50.7 mm). In 2018, the month with the greatest rainfall was August, with 87.5 mm, whereas only 62.9 mm was recorded in this month in 2017. In the maize growing season, the mean temperature was 19.1 °C in 2017 and 18.5 °C in 2018. The average daily temperature was above 20 °C for 81 days in 2017 and 73 days in 2018, respectively.

### Experimental design

The maize experiment was set up as a two-factorial randomized block design with three replications. Factor 1 involved plastic mulching (pm) or no plastic mulching (nm), while Factor 2 included sheep manure at three application rates (SM12, 1.2 t ha$^{-1}$, SM24, 2.4 t ha$^{-1}$, and SM36, 3.6 t ha$^{-1}$), resulting in a total of six treatments: SM12 nm, SM12 pm, SM24 nm, SM24 pm, SM36 nm, and SM36 pm. Each plot was 8 m long and 4 m wide. The

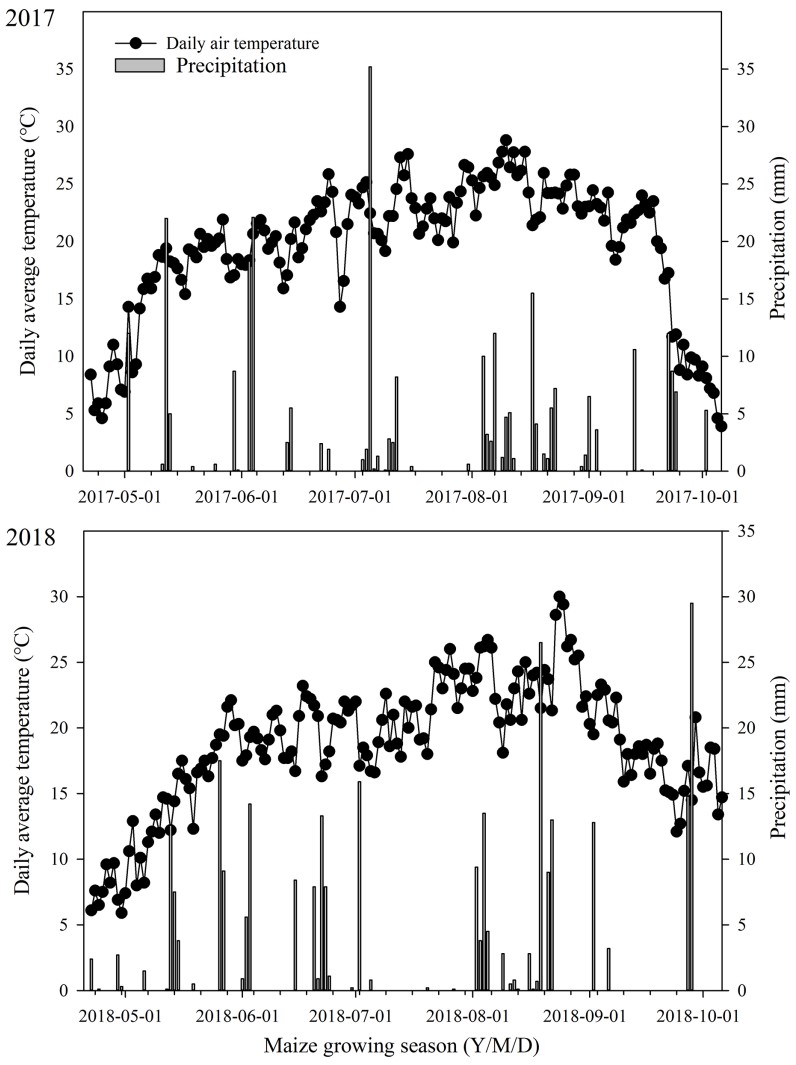

**Figure 1 Precipitation and daily average temperature during maize growing season.**

nutrient content of the manure was 0.75% N, 0.5% P and 0.45% K. Three weeks before sowing (2nd April 2017 and 5th April 2018), apply the organic fertilizer in a single application according to the specified amounts for different treatments. Spread the organic fertilizer evenly on the soil surface and plow it into the top 15–20 cm of the soil.

Maize (*Zea mays* L.), cv. 'Xianyu 335' was manually sown on 22nd April 2017 and 25th April 2018 and covered with a 2–3 cm soil layer. All plots followed a ridge-furrow pattern with narrow row spacing of 40 cm and wide row spacing of 70 cm, and a plant spacing of 35 cm within the rows. After the plot was prepared, the entire ridge surface in the three mulch treatments was covered with colourless, transparent, 0.008 mm thick polyethylene film plastic (Yonggu Suye Co., Ltd., Shaanxi, China) obtained from the local farmers' market. The planting density was 55,000 seeds ha$^{-1}$, with rows alternately wide (10-cm high × 70-cm wide) and narrow (15-cm high × 40-cm wide) consecutively. The in-row plant distance was 35 cm. The maize crop was harvested on 3rd October 2017 and 5th

October 2018. After harvest, the plastic film was collected for recycling by the manufacturer, and the soil was plowed to a depth of 25 cm. There was no irrigation throughout the growth period.

## Grain yield, biomass, and harvest index

At harvest, grain yield and aboveground biomass were measured on an area of 4 m$^2$ in the middle of each plot. The plants were divided into grains and stalks, which were weighed separately. All samples were dried for 30 min at 105 °C to rapidly stop plant metabolic activity and then dried to a constant weight at 70 °C to obtain the dry matter of the aboveground biomass (kg ha$^{-1}$). The harvest index was determined by the ratio of grain yield to aboveground biomass yield.

## Soil water content, evapotranspiration, and water use efficiency

During each growing season, soil moisture was measured using the gravimetric method on June 1, 2017, September 28, 2017, June 5, 2018, and September 25, 2018, at 20-cm depth increments (0 to 100 cm). In each plot, five soil cores (0–20 cm deep) were collected, with three taken from the centre of the ridge and two from the centre of the furrow, and combined into a single composite sample. The samples were air-dried, homogenized, and manually ground to pass through a 2-mm sieve for subsequent analysis of soil microbial community, available nitrogen, and enzymatic activity.

Nitrogen uptake for grain yield (NUG) was determined by multiplying the total grain dry matter accumulation by the nitrogen concentration. The evapotranspiration (ET, mm) as the crop water consumption was estimated using the following equation (*Wang et al., 2018a*):

$$ET = I + P + Cr + Dp \pm F \pm \Delta S \tag{1}$$

where $I$ is the amount of irrigation, which was zero since the experiment was conducted without any irrigation treatment; $P$ is the total rainfall; $Cr$ is the runoff, due to the installation of water-retaining facilities around the field, no significant rainfall events occurred throughout the entire growth period; $Dp$ is the deep seepage, which was negligible because the groundwater table was located at a depth of 80 m below the earth's surface; $F$ represents groundwater recharge, which was also negligible; $\Delta S$ is the change in soil moisture from the beginning to the end of the experiment. The contributions of groundwater recharge, runoff and deep seepage were negligible. The water use efficiency (WUE) was determined using the following equation (*Peng et al., 2023*):

$$WUE = \frac{GY}{ET}. \tag{2}$$

The equation for grain yield (GY) in kilograms per hectare (kg ha$^{-1}$) is determined by the evapotranspiration (ET) in millimetres (mm).

## Soil nitrate-N content and nitrogen use efficiency

In each plot soil samples were taken at five observation points (−30, −15, 0, 15, and 30 cm from the furrows and ridges) in 20 cm increments down to a depth of 60 cm as most of the

maize roots were found in this soil layer. Soil extraction was done with a 0.5 N potassium chloride solution at soil: liquid ratio of 1:3. A spectrophotometer was used to measure the soil nitrate-N content (UV-VIS 8500II; Xinmao Instrument Co., Ltd, Shanghai, China). Total nitrogen content of the maize samples was determined by the micro-Kjeldahl method. The majority of the maize roots were found in the 0–100 cm soil layer; therefore, soil samples were collected at a depth of 0–100 cm. Nitrogen uptake by plants was calculated by multiplying the dry matter weight of grains and stalks by their nitrogen concentration. Nitrogen use efficiency (NUE, in %) was analyzed using the following equation (*Wang et al., 2018c*):

$$NUE = \frac{N_{uptake, \ gy}}{N_{input}} \times 100\% \tag{3}$$

where $N_{uptake, \ gy}$ is the nitrogen fertilizer application level (kg ha$^{-1}$), and $N_{input}$ is the nitrogen uptake in the grain yield (kg ha$^{-1}$).

### Principal component analysis of yields, WUE, and NUE of maize

Principal component analysis (PCA) is a generic term for a technique that uses complex basic mathematical principles to convert a large number of potentially related variables into a smaller number of variables called principal components (*Wang et al., 2020*). PCA originated from multivariate data analysis. In general, PCA uses vector space transformation to reduce the dimensions of large data sets. Using mathematical projections, the original data set may involve many variables and can often be interpreted as a few variables (principal components). Therefore, in general, reviewing the dimensionality reduction data set will enable the user to discover trends, patterns, and outliers in the data, which is much easier than performing principal component analysis. Grain yield, biomass, harvest index, water use efficiency, grain N uptake, nitrogen use efficiency, and other indexes were selected for this study.

### Statistical analysis

Analysis of variance was conducted for grain yield, above ground biomass, harvest index, and water use efficiency using SPSS 16 (SPSS, Inc., Chicago, IL, USA). Differences between all treatments were detected using least significant difference (LSD) testing at the 0.05 significance level. Tukey's-b multiple range tests were used for paired mean comparisons at a 0.05 probability level.

## RESULTS

### Grain yield, biomass, and harvest index

The individual factors of mulching and the application rate of sheep manure significantly influenced maize grain yield, with no significant two-way interaction observed between these factors in both years (Table 1). The highest grain yield was recorded in the SM36 pm treatment, with yields of 8,296 kg ha$^{-1}$ in 2017 and 8,422 kg ha$^{-1}$ in 2018, which were significantly greater than those from SM12 nm (44% higher in 2017 and 41.5% higher in 2018), SM12 pm (31.3% higher in 2017 and 30.8% higher in 2018), SM24 nm (22.9%

**Table 1 Effect of organic N fertilizer and mulching on maize grain yield, biomass, and harvest index.**

| Years | Treatment | Sheep manure rate (t ha$^{-1}$) | Mulching | Grain yield (kg ha$^{-1}$) | Biomass (kg ha$^{-1}$) | Harvest index |
|---|---|---|---|---|---|---|
| 2017 | SM12 nm | 1.2 | No | 5,760 + 260 e | 18,687 + 1,245 b | 0.309 + 0.014 b |
| | SM12 pm | 1.2 | Yes | 6,320 + 347 de | 20,531 + 2,579 ab | 0.310 + 0.021 b |
| | SM24 nm | 2.4 | No | 6,752 + 340 cd | 22,032 + 2,320 ab | 0.308 + 0.018 b |
| | SM24 pm | 2.4 | Yes | 7,465 + 278 bc | 22,759 + 1,501 ab | 0.329 + 0.013 ab |
| | SM36 nm | 3.6 | No | 7,667 + 350 ab | 23,272 + 1,425 ab | 0.330 + 0.009 ab |
| | SM36 pm | 3.6 | Yes | 8,296 + 311 a | 23,837 + 1,625 a | 0.349 + 0.013 a |
| F | | | | <0.001 | 0.009 | 0.011 |
| M | | | | 0.001 | 0.253 | 0.074 |
| F × M | | | | 0.917 | 0.810 | 0.452 |
| 2018 | SM12 nm | 1.2 | No | 5,952 + 470 d | 18,469 + 15,759 c | 0.322 + 0.008 b |
| | SM12 pm | 1.2 | Yes | 6,437 + 298 cd | 19,966 + 1,476 bc | 0.323 + 0.009 ab |
| | SM24 nm | 2.4 | No | 6,881 + 496 bcd | 22,432 + 1,361 ab | 0.307 + 0.007 ab |
| | SM24 pm | 2.4 | Yes | 7,482 + 410 abc | 22,608 + 2,039 ab | 0.332 + 0.013 ab |
| | SM36 nm | 3.6 | No | 7,681 + 534 ab | 23,928 + 1,481 ab | 0.321 + 0.020 ab |
| | SM36 pm | 3.6 | Yes | 8,422 + 380 a | 24,203 + 1,260 a | 0.348 + 0.013 a |
| F | | | | <0.001 | 0.001 | 0.117 |
| M | | | | 0.012 | 0.392 | 0.011 |
| F × M | | | | 0.881 | 0.720 | 0.173 |

**Note:**
Values followed by the same letter(s) do not significantly differ at the 5% level of significance.

higher in 2017 and 22.4% higher in 2018), and SM24 pm (11.1% higher in 2017 and 12.6% higher in 2018). When averaged across all organic nitrogen fertilizer application rates, the grain yield for the mulching treatment was significantly higher than that of the no mulching treatment in both 2017 and 2018 (Table 1). The grain yield for the 3.6 t ha$^{-1}$ SM treatment averaged 8,016 kg ha$^{-1}$, which was 31% and 12.2% higher than the yields for the 1.2 and 2.4 t ha$^{-1}$ SM treatments, respectively (Table 1).

The rate of SM application significantly influenced the aboveground biomass of maize (Table 1). However, the individual factors of mulching and the interaction between mulching and soil moisture application rate did not have a significant effect on maize biomass (Table 1). The highest biomass was recorded with the SM36 pm treatment, which exhibited increases of 27.6% and 31% compared to the SM12 nm treatment in 2017 and 2018, respectively (Table 1). In comparison to the no mulching treatment, mulching resulted in a biomass increase of 4.7% in 2017 and 3% in 2018 (Table 1). When averaged across all mulching treatments and years, the biomass for the 3.6 t ha$^{-1}$ soil moisture treatment (23,810 kg ha$^{-1}$) was the highest, being 22.7% greater than that for the 1.2 t ha$^{-1}$ soil moisture treatment and 6% greater than that for the 2.4 t ha$^{-1}$ SM treatment, respectively (Table 1).

The individual factors of mulching (2017) and SM application rate (2018) significantly affected the harvest index, but there was no significant two-way interaction between mulch and SM application rate (Table 1). When averaged across all mulching treatments, the harvest index for the 3.6 t ha$^{-1}$ SM application was the highest, exceeding the harvest index

for the 1.2 and 2.4 t ha$^{-1}$ SM applications by 9.7% and 6.6%, respectively, in 2017 (Table 1). In 2018, the harvest index for mulching was 5.2% higher than that for the no-mulching treatments when averaged across all organic nitrogen fertilizer application rates (Table 1).

## Soil water content

The soil water content in the upper soil layer (20–60 cm) was less than that in the deep soil layer (60–100 cm) (Fig. 2). Averaged over all SM application rates and mulch treatments, the soil water content in the upper soil layer (0–40 cm) was 12.6%, 19.4%, 13.6%, and 14.4% lower than that in the deep soil layer (60–100 cm) on June 1, 2017 (Fig. 1A), September 28, 2017 (Fig. 1B), June 5, 2018 (Fig. 1C), and September 25, 2018 (Fig. 1D), respectively.

The soil water content was higher in the mulching treatments compared to the no mulching treatments. Specifically, on June 1, 2017, the soil water content was 16.1% higher in SM12 pm than in SM12 nm. In addition, it was 6.4% higher in SM24 pm and 4.9% higher in SM36 pm compared to SM24 nm and SM36 nm, respectively (Fig. 2A). The soil water content in the SM12 pm, SM24 pm, and SM36 pm treatments was 9.5%, 8.3%, and 10.6% higher than that in the SM12 nm, SM24 nm, and SM36 nm treatments on September 28, 2017, respectively (Fig. 2B); 9.1%, 12.6%, and 11.6% higher on June 5, 2018, respectively (Fig. 2C); and 11.3%, 8.9%, and 9.2% higher on September 25, 2018, respectively (Fig. 2D).

## Soil nitrate-N content

The soil nitrate-N content was higher in the early growth stages (June 1, 2017 and June 5, 2018) than in the later growth stages (September 28, 2017 and September 25, 2018) (Fig. 3). Averaged over all SM application rates and mulch treatments, the soil nitrate-N content in the early growth stage was 26.8% higher in 2017 and 28.7% higher in 2018 compared to the later growth stages (Fig. 3). The soil nitrate-N content in the treatments with mulching was 31.8%, 11.2%, and 1.8% higher on June 1, 2017, and 33.8%, 10.1%, and 12.3% higher on June 5, 2018, compared to the treatments with no mulching (Fig. 3). However, on September 28, 2017, the soil nitrate-N content in the SM24 pm and SM36 pm was 19.8% and 32.3% lower, respectively, than that in the SM24 nm and SM36 nm (Fig. 3). Averaged over all mulching treatments, the soil nitrate-N content for 1.2 t ha$^{-1}$ SM was the lowest, being 23.8% and 43.1% lower than the soil nitrate-N content for 2.4 and 3.6 t ha$^{-1}$ SM on June 1, 2017, respectively (Fig. 3). Averaged over all soil depths and mulching treatments, the soil nitrate-N content for 1.2 t ha$^{-1}$ SM was the highest on September 28, 2017. This content was 31.7% and 34.6% lower than the soil nitrate-N content for 2.4 SM and 3.6 t ha$^{-1}$ SM, respectively (Fig. 3).

Averaged over all soil depths, the soil nitrate-N content on June 1, 2017 was 18.3%, 11.4%, 13.7%, 8.1%, 12.4% and 16.3% higher than that on June 5, 2018 in SM12 nm, SM12 pm, SM24 nm, SM24 pm, SM36 nm and SM36 pm, respectively (Fig. 3). Averaged over all soil depths, the soil nitrate-N content on September 28, 2017 was 17.1%, 20.5%, 15.1%, 13.6%, 6.9% and 19.6% higher than that on September 25, 2018 in SM12 nm, SM12 pm, SM24 nm, SM24 pm, SM36 nm and SM36 pm, respectively (Fig. 3). When averaged over

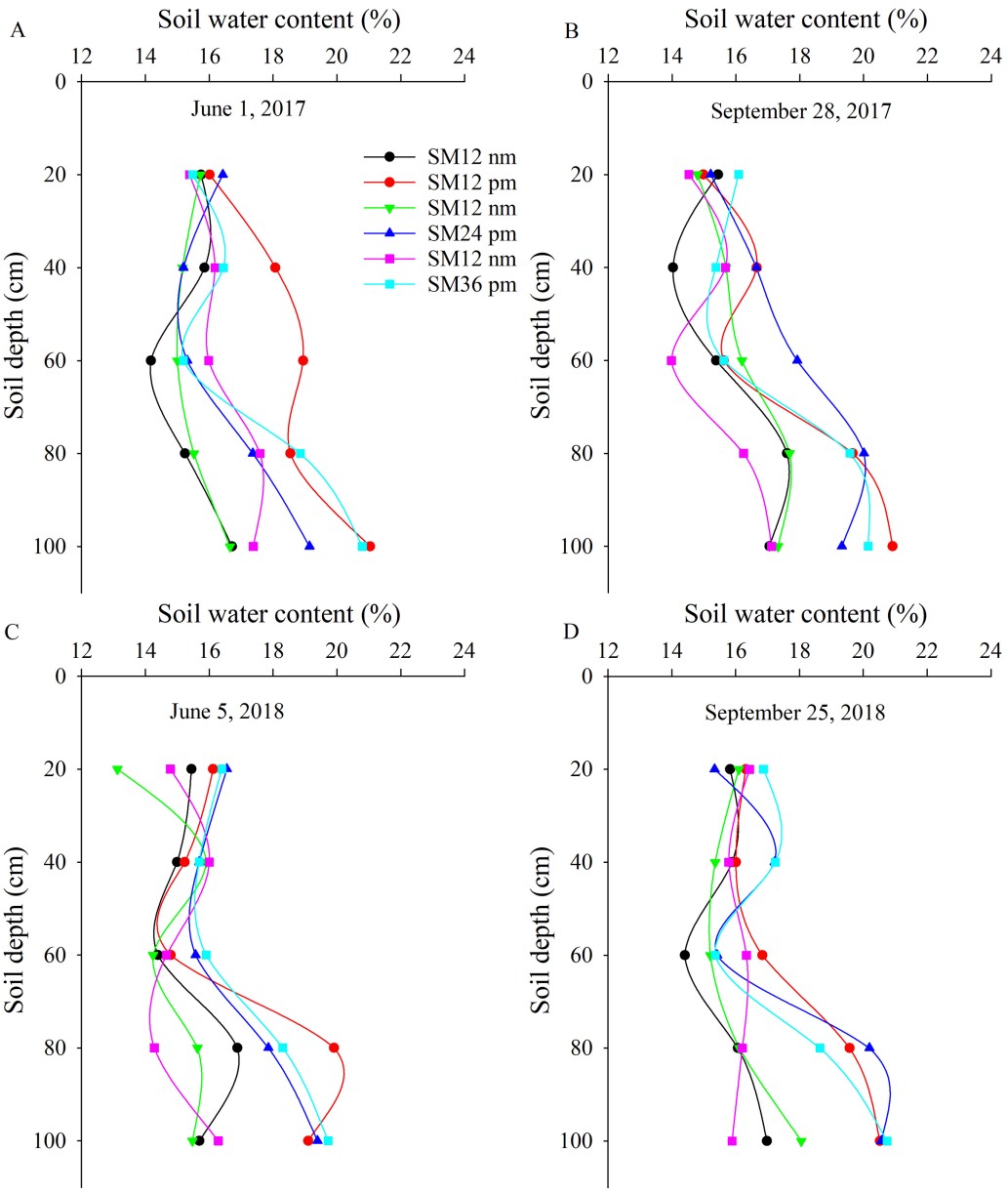

**Figure 2 Effect of organic N fertilizer and mulching on soil water content.** Note: SM12 nm, 1.2 t ha$^{-1}$ SM with no-mulching treatment; SM12 pm, 1.2 t ha$^{-1}$ SM with plastic mulching treatment; SM24 nm, 2.4 t ha$^{-1}$ SM with no-mulching treatment; SM24 pm, 2.4 t ha$^{-1}$ SM with plastic mulching treatment; SM36 nm, 3.6 t ha$^{-1}$ SM with no-mulching treatment; and SM36 pm, 3.6 t ha$^{-1}$ SM with plastic mulching treatment.

all soil depths and years, the soil nitrate-N content in the mulching treatment at 1.2 t ha$^{-1}$ SM was 31.8% higher than that in the no mulching treatment (Fig. 3). Similarly, when averaged over all soil depths and years, the soil nitrate-N content in the mulching treatment at 2.4 t ha$^{-1}$ SM was 12.4% higher than that in the no mulching treatment (Fig. 3). Averaged over all soil depths and years, the soil nitrate-N content in the mulching treatment was 10% higher than that in the no mulching treatment at 3.6 t ha$^{-1}$ SM (Fig. 3), indicating a significant difference.

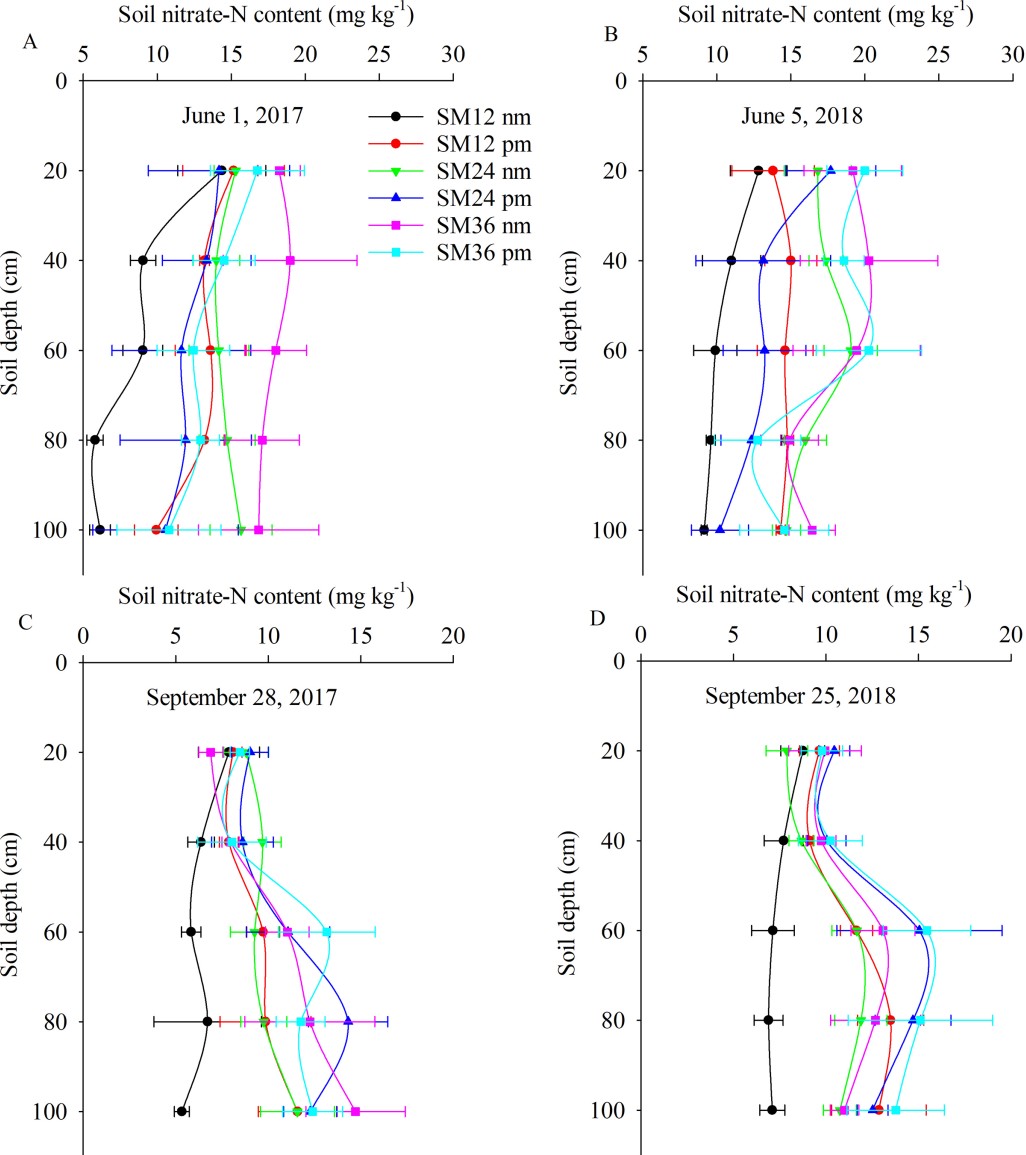

**Figure 3 Effect of SM application and mulching on soil nitrate-N content (mg kg$^{-1}$) at early and later growth stage.** Note: June 1, 2017 (A), June 5, 2018 (B), September 28, 2017 (C), September 28, 2017 (D). SM12 nm, B, SM12 pm, C, SM24 nm, D, SM24 pm, E, SM36 nm, F, SM36 pm; G, SM12 nm, H, SM12 pm, I, SM24 nm, J, SM24 pm, K, SM36 nm, L, SM36 pm.SM12 nm, 1.2 t ha$^{-1}$ SM with no-mulching treatment; SM12 pm, 1.2 t ha$^{-1}$ SM with plastic mulching treatment; SM24 nm, 2.4 t ha$^{-1}$ SM with no-mulching treatment; SM24 pm, 2.4 t ha$^{-1}$ SM with plastic mulching treatment; SM36 nm, 3.6 t ha$^{-1}$ SM with no-mulching treatment; and SM36 pm, 3.6 t ha$^{-1}$ SM with plastic mulching treatment.

## Water use efficiency, nitrogen uptake in grain yield, and nitrogen use efficiency

The individual factors of mulching and SM application rate significantly affected WUE, and there was a significant interaction between mulching and organic N fertilizer application rate on WUE (Table 2). The WUE ranged from 1.95 to 2.41 kg m$^{-3}$ in 2017 and

**Table 2 The effect of fertilizer and mulching on water use efficiency, nitrogen uptake in grain yield, and nitrogen use efficiency.**

| Years | Treatment | Sheep manure rate (t ha$^{-1}$) | Mulching | WUE (kg m$^{-3}$) | NUG (kg ha$^{-1}$) | NUE (%) |
|---|---|---|---|---|---|---|
| 2017 | SM12 nm | 1.2 | No | 2.01 + 0.02 c | 33.21 + 2.40 c | 36.90 + 2.67 b |
| | SM12 pm | 1.2 | Yes | 1.95 + 0.06 c | 37.09 + 2.65 c | 41.21 + 2.94 a |
| | SM24 nm | 2.4 | No | 2.04 + 0.03 c | 63.21 + 1.42 b | 35.11 + 0.79 b |
| | SM24 pm | 2.4 | Yes | 2.17 + 0.06 b | 65.80 + 2.60 b | 36.56 + 1.45 b |
| | SM36 nm | 3.6 | No | 2.31 + 0.06 a | 78.05 + 2.51 a | 28.91 + 0.93 c |
| | SM36 pm | 3.6 | Yes | 2.41 + 0.06 a | 81.71 + 1.50 a | 30.26 + 0.55 c |
| F | | | | <0.001 | <0.001 | <0.001 |
| M | | | | 0.039 | 0.392 | 0.566 |
| F × M | | | | 0.016 | 0.028 | 0.027 |
| 2018 | SM12 nm | 1.2 | No | 2.05 + 0.11 b | 37.05 + 2.43 c | 41.16 + 2.70 a |
| | SM12 pm | 1.2 | Yes | 2.04 + 0.06 b | 40.00 + 2.63 c | 44.45 + 2.92 a |
| | SM24 nm | 2.4 | No | 2.07 + 0.12 b | 65.37 + 2.93 b | 36.32 + 1.63 b |
| | SM24 pm | 2.4 | Yes | 2.10 + 0.08 b | 64.41 + 3.52 b | 35.78 + 1.95 b |
| | SM36 nm | 3.6 | No | 2.17 + 0.09 b | 77.95 + 4.49 a | 28.87 + 1.66 c |
| | SM36 pm | 3.6 | Yes | 2.44 + 0.07 a | 79.12 + 3.18 a | 29.30 + 1.18 c |
| F | | | | 0.001 | <0.001 | <0.001 |
| M | | | | 0.046 | 0.505 | 0.304 |
| F × M | | | | 0.047 | 0.596 | 0.298 |

**Note:**
Values followed by the same letter(s) do not show significant differences at the 5% level of significance. WUE represents water use efficiency (kg m$^{-3}$); NUG represents nitrogen uptake in grain yield (kg ha$^{-1}$); NUE represents nitrogen use efficiency (%); SM12 nm represents 1.2 t ha$^{-1}$ SM with no-mulching treatments; SM12 pm represents 1.2 t ha$^{-1}$ SM with plastic mulching treatments; SM24 nm represents 2.4 t ha$^{-1}$ SM with no-mulching treatments; SM24 pm represents 2.4 t ha$^{-1}$ SM with plastic mulching treatments; SM36 nm represents 3.6 t ha$^{-1}$ SM with no-mulching treatments; SM36 pm represents 3.6 t ha$^{-1}$ SM with plastic mulching treatments; F represents fertilizer; M represents mulching. $n$ = 3.

from 2.04 to 2.44 kg m$^{-3}$ in 2018 among the different treatment groups (Table 2). The WUE in the mulching treatment was 3% and 7% higher than that in the no mulching treatment in 2017 and 2018, respectively. Averaged over all mulching treatments, the WUE for 1.2 t ha$^{-1}$ SM was the highest, which was 9% and 32% higher than the WUE for 2.4 and 3.6 t ha$^{-1}$ SM in 2017, respectively (Table 2). When averaged over all mulching treatments, the WUE for 3.6 t ha$^{-1}$ SM was the highest, at 18.7%, which was 32% lower than the WUE for 2.4 and 1.2 t ha$^{-1}$ SM in 2017, respectively (Table 2).

Mulching as an individual factor significantly affected nitrogen uptake in grain yield, and there was a significant interaction between mulching and SM application rate on nitrogen uptake in grain yield in 2017 (Table 2). Averaged over all SM application rates, the nitrogen uptake in grain yield with mulching treatments was only 5.8% and 1.8% higher than that in 2017 and 2018, respectively (Table 2). Averaged over all mulching treatments, the nitrogen uptake in grain yield for 3.6 t ha$^{-1}$ SM was the highest in both 2017 and 2018, with increases of 23.8% and 127.3% compared to 2.4 t ha$^{-1}$ SM, and 21% and 103.9% compared to 1.2 t ha$^{-1}$ SM, respectively (Table 2). Compared with 2018, the nitrogen uptake in grain yield was 1.3% higher than that in 2017 on average over all mulching and organic N fertilizer application rate treatments (Table 2).

Mulching as an individual factor significantly affected NUE, and there was a significant interaction between mulching and SM application rate on NUE in 2017 (Table 2). Averaged over all mulching treatments, the NUE for 3.6 t ha$^{-1}$ SM was the lowest, which was 9% and 32% lower than the NUE for 2.4 and 1.2 t ha$^{-1}$ SM in 2017, respectively (Table 2). Averaged over all mulching treatments, the NUE for 3.6 t ha$^{-1}$ SM was the lowest, which was 18.9% and 46.4% lower than the NUE for 2.4 and 1.2 t ha$^{-1}$ SM in 2018, respectively (Table 2). Compared with 2018, the nitrogen uptake in grain yield was 3.3% higher than that in 2017 on average over all mulching and organic N fertilizer application rate treatments (Table 2).

## Principal component analysis and correlation analysis

Significant correlations were observed between grain yield, biomass, harvest index, water use efficiency, nitrogen uptake in grain yield, nitrogen use efficiency, and soil nitrate-N content at the early growth stage (SNCE), with correlation coefficients ranging from 0.777 to 0.948. In contrast, soil water content did not exhibit a correlation with these indicators (Tables S1). Additionally, NUE showed a negative correlation with grain yield, biomass, water use efficiency, and nitrogen uptake by grain, with correlation coefficients ranging from 0.762 to 0.886 (Tables S1).

Measurements of grain yield, biomass, harvest index, water use efficiency, nitrogen uptake in grain yield, nitrogen use efficiency, soil water content at the early growth stage, soil water content at the late growth stage, soil nitrate-N content at the early growth stage, and soil nitrate-N content at the late growth stage were clustered together. The mean values of the two consecutive growing seasons during 2017 and 2018 were calculated (Tables S2 and S3). These values were used in the following steps: (1) the data was converted to a standardized value (Tables S4 and S5), (2) the correlation matrix was calculated from normalized values (Tables S6 and S7), and (3) the total variance, eigenvalue of contribution rate, and contribution rate accumulation were obtained by principal component analysis (Tables S8 and S9). In this study, three components were extracted from the matrix of composite indicators (Tables S10 and S11). The comprehensive quality rankings using principal component analysis are shown in Fig. 4.

## DISCUSSION

Mulching and SM treatments increased maize yields by 8.2–44% compared to no-mulching treatments. This phenomenon may be related to early soil temperature and humidity increases in cold semiarid areas of China (Li et al., 2013). Another reason may be that the yield increase effect of organic fertilizer is not as significant as that of mineral fertilizer (Shah et al., 2023). This study shows that grain yield for 2017 was lower than that in 2018 on average over all factors. We suspect that the regulation of soil temperature by mulch is directly coupled with the effect of organic fertilizers, so it is necessary to further study the effect of organic fertilizers on soil temperature. Our results indicated a significant effect on maize biomass, but mulching as a single factor and the interaction of mulching and organic fertilizer application rate had no significant influence on aboveground biomass. Plastic film mulch significantly increased the aboveground biomass by 15% in the

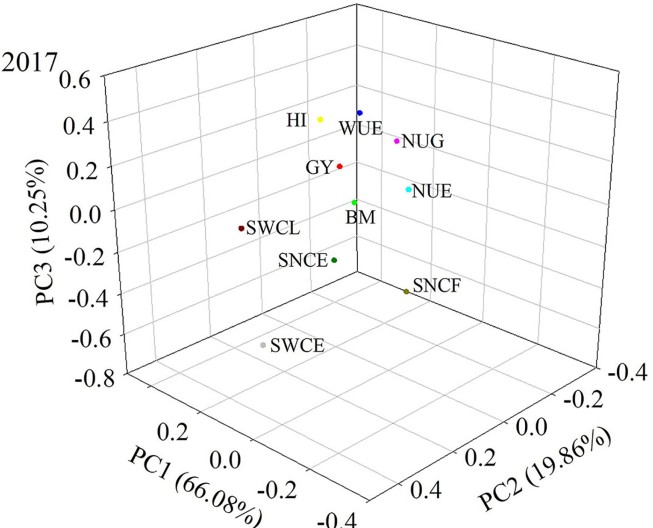
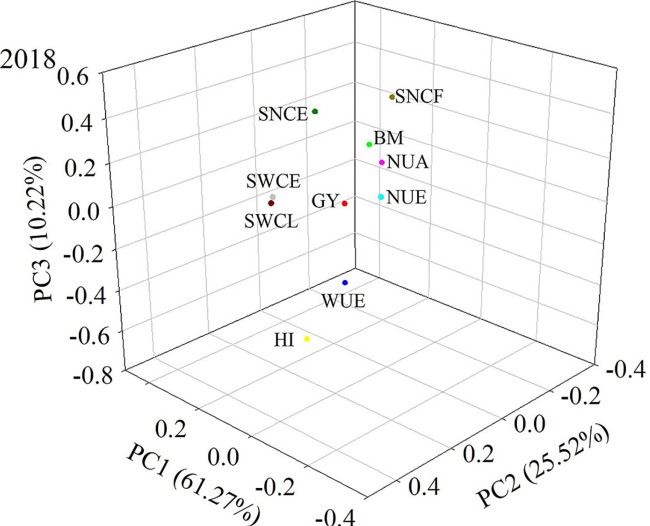

**Figure 4 Principal component analysis of all indicators using the average values in both years.** Note: GY (grain yield), BM (biomass), HI (harvest index), WUE (water use efficiency), NUG (nitrogen uptake in grain yield), NUE (nitrogen use efficiency), SWCE (soil water content at early growth stage), SWCL (soil water content at late growth stage), SNCE (soil nitrate-N content at early growth stage), SNCL (soil nitrate-N content at late growth stage).

warmer year and by 52% in the drier year (*Wang et al., 2016*). This phenomenon may be related to maize varieties and planting density (*Wang et al., 2018b*). Our results showed that mulching improved the stability of maize grain yield under different fertilizer rates but did not significantly alter biomass. Particularly, there was no significant difference in grain yield, biomass, WUE and NUE between SM36 nm and SM36 pm under 36 t ha$^{-1}$ sheep manure with or without mulching in this study.

The individual factors associated with mulching and organic fertilizer application rate significantly affected the harvest index of maize. The harvest index increased with an increase in the organic N fertilizer application rate, and the harvest index for mulching was higher than that in the no mulching treatments. An important reason for the effect of organic fertilizer application on maize harvest index may be the low nitrate N content around the root zone, reflecting the fact that a large amount of N is absorbed during the grain filling stage (*Zhang et al., 2009*). Researchers reported that adequate basal nitrogen fertilizer provided sufficient soil nitrate-N for maize growth before the grain filling period, and that this absorbed N was used for re-translocation from vegetative organs to grains during the grain filling period (*Chung et al., 2000*).

Nitrogen is the most sensitive element to crop growth and development, and it plays a vital role in increasing crop yield and improving nitrogen use efficiency (*Lyu et al., 2024*). The contents of nitrate, alkali-hydrolysable nitrogen, and total nitrogen in soil can reflect soil fertility (*Li et al., 2024c*). The addition of organic fertilizer can significantly increase soil porosity, rainfall infiltration rate and soil water storage capacity (*Li et al., 2024b*). In this study, mulching and organic fertilizer rate significantly affected the water use efficiency in 2017 and 2018. Some studies reported that high mineral fertilizer application rates increased the transpiration and evapotranspiration of fields and aggravated the

consumption of soil water so that the rainfall in the period between crops could not effectively replenish the water consumed in the growing season (Kang et al., 2017). In our study, we found that the water consumed by the crops in the growing season can be effectively replenished by rainfall, and sometimes even the storage of rainfall is greater than the water consumption of the crops. This may be related to the imbalance of interannual rainfall and the application of organic fertilizers, so it is necessary to study the mechanism of organic fertilizers' influence on soil water content.

This study showed that there was a significant correlation between yield and soil nitrate-N content but no significant correlation between yield and soil water content in early and later growth stage. We determined the correlation between soil nitrogen content and yield at harvest time, which indicated that there was a significant correlation between residual soil nitrogen and yield. Increasing the use of soil nitrogen through appropriate nitrogen fertilizer application rates can improve the overall nitrogen use efficiency thereby reducing nitrogen crop production costs and unwanted losses (Ullah et al., 2019). Studies have shown that higher soil nitrogen levels lead to higher soil nitrate leaching, increased demand of crops for fertilizer, and finally resulting in lower NUE (Jia et al., 2014). For this reason, many experiments have been conducted to determine the optimal nitrogen application rate (Ciampitti et al., 2013; Li et al., 2020). However, with increasing nitrogen application rate the corresponding grain yield increase flattens out (Guan et al., 2014). In this study, there was no significant difference between the application of organic fertilizer at 2.4 and 3.6 t ha$^{-1}$ SM.

## CONCLUSION

The soil water content was higher in the mulching treatment compared to the non-mulching treatment. Averaged across all organic nitrogen fertilizer application rates and mulch treatments, the soil nitrate-N content during the early growth stage was 26.8% and 28.7% higher in 2017 and 2018, respectively. A significant interaction was observed between mulching and the organic nitrogen fertilizer application rate concerning WUE. When averaged across all mulching treatments, the NUE for 3.6 t ha$^{-1}$ of SM was the lowest, being 19.3% and 32.1% lower than the NUE for 2.4 and 1.2 t ha$^{-1}$ SM in 2018, respectively. These results suggest that the application of organic fertilizer at a rate of 3.6 t ha$^{-1}$ SM in conjunction with mulching is a high-priority, recommended treatment for the Loess Plateau areas.

### Funding

This research was funded by the Shanxi Province Key Special Project for the Fusion of "Two Chains", Grant No. 2023LLRH-01; Shaanxi Provincial Department of Education Youth innovation team construction research project, Grant No. 21JP141, 22JP101, 23JP189. The funders had no role in study design, data collection and analysis, decision to publish, or preparation of the manuscript.

## Grant Disclosures

The following grant information was disclosed by the authors:
Shanxi Province Key Special Project for the Fusion of "Two Chains": 2023LLRH-01.
Shaanxi Provincial Department of Education Youth Innovation Team Construction
Research Project: 21JP141, 22JP101, 23JP189.

## Competing Interests

Xiukang Wang is an Academic Editor for PeerJ.

## Author Contributions

- Yingying Xing conceived and designed the experiments, performed the experiments, analyzed the data, authored or reviewed drafts of the article, and approved the final draft.
- Jintao Fu performed the experiments, authored or reviewed drafts of the article, and approved the final draft.
- Xiukang Wang conceived and designed the experiments, analyzed the data, prepared figures and/or tables, authored or reviewed drafts of the article, and approved the final draft.

## Data Availability

The raw measurements are available in the Supplemental File.

## Supplemental Information

Supplemental information for this article can be found online at http://dx.doi.org/10.7717/peerj.18644#supplemental-information.

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
