# Peer review of "Effect of mulching and organic manure on maize yield, water, and nitrogen use efficiency in the Loess Plateau of China"

_PeerJ, doi:10.7717/peerj.18644_

## Round 0.1 · original submission · Major Revisions

We have concluded the revision of your manuscript. Three expert reviewers have provided their evaluations and their comments and suggestions are given below. Due to the extent of the comments I believe major revisions are in order. As you prepare your new manuscript please ensure that the revision includes a careful accounting of every suggestion and how (and where) it was incorporated or otherwise addressed in the manuscript so that the reviewers and I have the easiest time verifying the improvements and clarifications.

·

Basic reporting

Abstract
L23: nonrenewable should be written as non-renewable
Introduction
L43: has merely should be written as have merely
L107: maize yield and water and fertilizer use efficiency to be re-written as maize yield, water and fertilizer use efficiency……..
L274: The WUE ranged from 1.95 to 2.41 kg m-3 in 2017. Write the unit in superscript m-3
L275: Write the unit in superscript (m-3)

Experimental design

No Comment

Validity of the findings

No Comment

Additional comments

No Comment

Reviewer 2 ·

Basic reporting

(1) English writing needs to be improved;

(2) Redundant and unnecessary details existed in Abstract (lines 20-26), Introduction (which is too long). Please refine;

(3) Lines 56-58: Please provide details about the research progress with citations to support the statement.

(4) Figure 3: An error was found on legend SM36 nm (the last one should be SM 36 pm); partial data in fig.3D were missed;

(5) Statistically significant letters or asterisks should be added in Figures 3 and 4;

(6) Figures 3 and 4: it is a bit difficult to compare differences in data by colors and shape of points, which are ambiguous. I suggest to adjust the type of lines to make them clear;

(7) The manuscript has a clear hypothesis.

Experimental design

(1) Line 131: Details about the application timing and method of organic fertilization should be clearly stated.

Validity of the findings

(1) In Results, all the data comparison should be statistically sound (i.e., were they statistically significant?). Pleased revise.

(2) Why the authors only measured nitrate nitrogen in soils, especially when calculating the NUE? Other types of soil nitrogen (e.g., TN) should also be investigated.

(3) The current dataset is too simple, which only includes data related to yield, soil nitrate-N and moisture content, weather data, etc. I suggest the authors providing more data (related to crop physiological traits and different types of soil nitrogen) to make the conclusions more robust.

(4) Lines 254: the soil nitrate-N content for 1.2 t ha-1 SM was the highest or the lowest?

Additional comments

The current study is a little bit superficial, the findings and conclusion are general, as I believe that many similar studies have done before regarding the effects from mulching and organic fertilization. Although the authors stated that limited studies were conducted in the Loess Plateau in China, however, recent research progress with citations are needed to support the current statement and emphasize the significance of the current study. Also, more data related to crop physiology and changes in content of different types of soil N are encouraged to provide, to make the conclusion more robust.

·

Basic reporting

In general, the study has clear objectives and was supported with a few data set and solution.
Please see my comments below:
Abstract
L27: Is it under randomized complete block design or factorial RCBD clearly state
Introduction
L83-84: how organic fertilizer helps to increase WUE and NUE justify with reference. add few lines.
L85-87: add references about those statement
Materials and methods
L143: the plant population seems to be mentioned wrongly as per spacing you have mentioned. Please recheck.
L156: how you have collected the soil sample for moisture measurement that should be mentioned.
L157: why you have waited 2-3 days after rain, state the reason
L163: In the equation have you consider total rainfall or effective rainfall? How you measure runoff and deep seepage
Clearly mentioned all the parameter of the equation what needs to be considered during estimation of ET with references
L173-174: what was the height of the ridges? The sentences clearly not describing the sampling details, rewrite the sentences in a simple language
Result
Well written but further need improvement particularly describing the mulch and manure interaction effect.
Why you have chosen to present only the result of infraction effect through all the two factors resulted significant variation in terms of different observed data
Modify the table accordingly or present the individual factors data in other presentable format
Why you have chosen only those two date for observing moisture and nitrate concentration? What is the importance in selecting those date?
Discussion
Need improvement a lot in this section and how your analysis like PCA or correlation matrix is justified your observed result that need to be highlighted.
Checked all the references

Experimental design

Recheck the design of the experiment which has been written in abstract and the test also

Validity of the findings

Its a good finding but need some further improvement

Additional comments

need few improvement in the manuscript

---

## Round 0.2 · Minor Revisions

Tw expert reviewers have evaluated your revised manuscript and their comments cab be seen below. While the reviewers are satisfied with the changes made to the manuscript, one of the reviewers is suggesting that figures 2 and 3 be revised and I agree that these should be improved.

Reviewer 2 ·

Basic reporting

No further comments.

Experimental design

No further comments.

Validity of the findings

No further comments.

Additional comments

Thank the authors for the responses and revisions. The quality of manuscript has been improved.

Minor comments:
Regarding the same treatments, consistent colors and the shape of points are highly recommended in figures 2 and 3. Moreover, the line colors of SM36 nm and SM36 pm are hard to differentiate in Fig. 3, please modify.

·

Basic reporting

No comment

Experimental design

no comments

Validity of the findings

no comment

Additional comments

Authors addressed all the quarries i had raised . I am satisfied with the revised manuscript.
Thanks

---

## Round 0.3 · accepted · Accept

I am satisfied with the changes made to the figures and feel that the manuscript is now ready for acceptance.